# Obesity Stigma: Is the ‘Food Addiction’ Label Feeding the Problem?

**DOI:** 10.3390/nu11092100

**Published:** 2019-09-04

**Authors:** Helen K. Ruddock, Michael Orwin, Emma J. Boyland, Elizabeth H. Evans, Charlotte A. Hardman

**Affiliations:** 1Department of Psychological Sciences, University of Liverpool, Liverpool L69 7ZX, UK; 2School of Psychology, University of Birmingham, Birmingham B15 2TT, UK; 3School of Psychology, Newcastle University, Newcastle NE1 7RU, UK

**Keywords:** food addiction, obesity, stigma, eating behavior, attitudes

## Abstract

Obesity is often attributed to an addiction to high-calorie foods. However, the effect of “food addiction” explanations on weight-related stigma remains unclear. In two online studies, participants (*n* = 439, *n* = 523, respectively, recruited from separate samples) read a vignette about a target female who was described as ‘very overweight’. Participants were randomly allocated to one of three conditions which differed in the information provided in the vignette: (1) in the “medical condition”, the target had been diagnosed with food addiction by her doctor; (2) in the “self-diagnosed condition”, the target believed herself to be a food addict; (3) in the control condition, there was no reference to food addiction. Participants then completed questionnaires measuring target-specific stigma (i.e., stigma towards the female described in the vignette), general stigma towards obesity (both studies), addiction-like eating behavior and causal beliefs about addiction (Study 2 only). In Study 1, participants in the medical and self-diagnosed food addiction conditions demonstrated greater target-specific stigma relative to the control condition. In Study 2, participants in the medical condition had greater target-specific stigma than the control condition but only those with low levels of addiction-like eating behavior. There was no effect of condition on general weight-based stigma in either study. These findings suggest that the food addiction label may increase stigmatizing attitudes towards a person with obesity, particularly within individuals with low levels of addiction-like eating behavior.

## 1. Introduction

According to recent statistics, more than one-third of the world’s population is overweight or obesity. In the UK, these rates are even higher, with 64% of adults classed as having overweight or obesity [1]. Despite its prevalence, people with obesity frequently experience devaluation and discrimination (known as weight-related stigma) within educational, workplace, and healthcare settings [2]. Evidence also suggests that people may be more likely to face discrimination because of their weight than because of their ethnicity, gender, or sexual orientation [3]. Weight-related stigma has negative consequences for individuals’ psychological and physical well-being [2,4,5] and may impede weight-loss by prompting maladaptive eating patterns and exercise avoidance [2]. 

Negative attitudes towards people with obesity can be exacerbated by beliefs about the *causes* of weight-gain. This is central to attribution theory, which suggests that people make judgements about the cause of a condition; in turn, these judgements determine their attitudes towards an individual [6,7]. For example, attributing obesity to factors that are within personal control (e.g., food choices) is thought to perpetuate obesity stigma [8]. Conversely, stigmatizing attitudes may be attenuated by the belief that weight-gain is caused by uncontrollable factors (e.g., genetics). In support of this, weight-related stigma was found to be most prevalent amongst individuals who believed that obesity was within personal control and caused by a lack of willpower, inactivity, and overeating [9,10]. Similar findings have been obtained from studies in which participants’ causal beliefs about obesity were experimentally manipulated. Specifically, participants who read an article that stated that obesity is caused by overeating and a lack of exercise demonstrated more stigmatizing attitudes than participants in a ‘no-prime’ control condition or those who read a neutral article about research into memory skills [11,12]. Conversely, participants who were led to believe that obesity is caused by physiological factors (i.e., factors that are beyond personal control) demonstrated less weight-related stigma than those in a control condition [8,13]. 

One increasingly prevalent etiological theory is that obesity is caused by an addiction to high-calorie foods [14]. Proponents of this idea suggest that food and drugs have similar effects on the brain and argue that the clinical symptoms of substance abuse coincide with the behaviors and experiences of people who engage in compulsive overeating [15,16]. While this idea is widely debated throughout the scientific community (e.g., [17,18,19]), the concept of food addiction has been readily accepted by the general public [20]. Indeed, research suggests that the majority of people believe that obesity can be caused by food addiction [21], and up to half of people believe that they are themselves addicted to food [22,23,24]. In light of its popularity, it is important to establish how food addiction models of obesity might affect weight-related stigma. 

A small number of studies have examined the effect of the food addiction label on obesity stigma. However, results to date have been inconsistent [25,26]. In one study [27], participants’ attitudes towards a person with ‘food addiction’ were compared with attitudes towards persons with obesity, drug addiction, and disability. The study reported similarly high levels of stigma towards the “obese” and “food addict” labels and, when combined, these labels together elicited greater stigma than either label alone. These findings align with those obtained by Lee et al. [21] who found that, while the majority (72%) of survey respondents believed that obesity could be caused by a ‘food addiction’, more than half held the view that people with obesity are responsible for their condition (which would be expected to perpetuate obesity stigma). However, in contrast, Latner et al. [28] found that providing a food addiction explanation for obesity appeared to *reduce* weight-stigma. In this study, participants read one of two descriptions of a woman with obesity. In one condition (i.e., the ‘food addiction’ condition), the woman was described as fitting “the typical profile of someone who is addicted to food”. In another condition (i.e., the ‘non-addiction’ condition), the woman was described as “someone who makes unhealthy food choices”. The study found that participants in the food addiction condition displayed lower levels of stigma towards the woman, and towards people with obesity more generally, compared with those in the non-addiction condition. 

Inconsistent findings in previous studies may be explained by differences in participants’ causal beliefs about food addiction. Specifically, the effect of the “food addiction” label on obesity stigma may depend on the extent to which food addiction is perceived to be a legitimate medical condition. One qualitative study found that people with overweight and obesity were reluctant to label themselves as a food addict due to concerns that this would be viewed as an ‘excuse’ for overeating [29]. Indeed, providing excuses for weight gain may exacerbate negative attitudes towards those with obesity [30]. In contrast, attributing obesity to a medically diagnosed ‘food addiction’ may legitimize the condition and help to reduce weight-related stigma by removing personal responsibility from the individual [31,32].

To test these ideas, across two studies, we examined the effect of medically-diagnosed and self-diagnosed food addiction on weight-related stigma. Using a similar technique to Latner et al. [28], participants read one of three vignettes which described a woman with obesity. In the ‘medical’ condition, the vignette stated that the woman had been diagnosed with food addiction by her general practitioner (GP). In the ‘self-diagnosed’ condition, the vignette stated that the woman believed herself to be a food addict. There was no reference to food addiction in the control condition. Subsequent attitudes towards the woman (i.e., target-specific stigma) and obesity in general (i.e., general stigma) were then assessed. We hypothesized that weight-related stigma would be significantly lower in the medical condition, and higher in the self-diagnosed condition, relative to in the control condition. Based on previous findings [28], we predicted that the food addiction label would influence both target-specific and general weight-related stigma.

## 2. Study 1 Method

### 2.1. Participants

Female participants were invited to take part in a study into ‘perceptions of employability among students’. Participants were recruited via social media advertisements and on internal webpages at the University of Liverpool, UK. Participants who were enrolled in the Psychology degree program at the University received course credits in exchange for taking part. A total of 440 participants completed the survey (533 participants started the study, but 93 did not complete all of the measures and so were excluded from analyses). To be eligible to take part, participants were required to be aged over 18 years old. The majority of participants were students (81%), and 90% of the sample were Caucasian. The mean age of participants was 21.2 y (SD = 7.1), and the mean self-reported body mass index (BMI) was 22.2 kg/m^2^ (SD = 3.4). Participants with a self-reported BMI over 30 kg/m^2^ (i.e., classified as having obesity) comprised 2.7% of the sample, 12.5% had a self-reported BMI between 25–29.9 kg/m^2^ (i.e., ‘overweight’), 76.8% had a self-reported BMI between 18.5–24.9 kg/m^2^ (i.e., healthy weight), and 8.0% had a BMI below 18.5 kg/m^2^ (i.e., ‘underweight’). Participants provided informed consent prior to completing the study. Ethical approval was granted by the University of Liverpool’s ethics committee (approval code: IPHS-1516-SMc-259-Generic RETH000619).

### 2.2. Procedure

The study was delivered via the online survey platform, Qualtrics (Qualtrics, Provo, UT, USA). Participants were asked to read an information sheet and, if they wished to continue with the study, were required to tick a consent box. On the first screen of the survey, a picture of a woman with obesity (“Paulina”) was displayed, along with a short vignette which described her hobbies, family, and education (see online Appendix A). Paulina was also described as being ‘very overweight’. Participants were randomly allocated to view one of three versions of the vignette: (1) In the ‘medical’ condition, the vignette stated that Paulina’s “GP had recently diagnosed her as having a food addiction”; (2) in the ‘self-diagnosed’ condition, the vignette stated that Paulina “believes herself to be addicted to food”; (3) in the ‘control’ condition, there was no mention of food addiction. After reading the vignette, participants completed the measures in the following order: Modified Fat-Phobia Scale (M-FPS) (to assess target-specific stigma towards Paulina), employability questionnaire (included as part of the cover story), Anti-fat Attitudes (AFA; to assess general stigma towards people with obesity), and the Dutch Eating Behavior Questionnaire (DEBQ; to assess external, restrained, and emotional eating behavior). Participants were then asked to indicate their gender, age, ethnicity, occupation, and height and weight (which were used to calculate BMI). They then completed the item about self-perceived food addiction. After completing the study, participants read a debrief sheet which explained the true aim of the study.

### 2.3. Measures

#### 2.3.1. Target Specific Stigma: Modified Fat-Phobia Scale (M-FPS)

The 14-item Fat Phobia Scale [33] was modified such that participants were asked to indicate their beliefs about a fictional individual named Paulina (Paulina was the name of the target female featured in the vignette. See Section 2.2). This scale consists of 14 pairs of antonyms which could be used to describe individuals with obesity (e.g., ‘lazy’ vs. ‘industrious’). Higher scores on the M-FPS (i.e., indicative of more negative attitudes) have been positively associated with beliefs that obesity is within personal control [9]. Participants were required to indicate their perceptions of Paulina by selecting one of five points between each pair of words. A mean score was calculated for each participant. Higher scores on this measure indicated more negative attitudes towards Paulina. In the current sample, the internal reliability of the M-FPS was high (Cronbach’s α = 0.834). 

#### 2.3.2. General Stigma: Anti-fat Attitudes (AFA)

The AFA [8] consists of 13 items which assess stigmatizing attitudes toward individuals with obesity (e.g., “I dislike people who are overweight or obese”). Responses are provided on a 9-point scale ranging from ‘Very strongly disagree’ to ‘Very strongly agree’ (in Study 1, a 5-point Likert scale was used, but this was corrected to a 9-point scale in Study 2). Higher scores indicate stronger anti-fat attitudes. The scale comprises three subscales which assess dislike (i.e., obesity stigma), willpower (i.e., beliefs about weight controllability), and fear of fat (i.e., concerns about personal weight gain) (Cronbach’s α = 0.796). 

#### 2.3.3. Dutch Eating Behavior Scale (DEBQ)

The DEBQ [34] consists of 33 items which assess eating behavior. The scale comprises three subscales assessing Restrained Eating (DEBQ-R; 10 items), Emotional Eating (DEBQ-EM; 13 items), and External Eating (DEBQ-EX; 10-items). Previous research has demonstrated the ability of the DEBQ to predict restrictive eating tendencies [35], eating in response to external food-cues [36], and stress-induced eating [37]. Responses are recorded on a 5-point Likert-type scale ranging from ‘Never’ to ‘Very often’. Higher scores indicate greater restrained, emotional, or external eating. The DEBQ was included to ensure that participants did not differ, between conditions, with regards to their eating behavior. The internal reliability for each of the subscales was high (DEBQ-R: Cronbach’s α = 0.933; DEBQ-EX: Cronbach’s α = 0.869; DEBQ-EM Cronbach’s α = 0.932).

#### 2.3.4. Self-Perceived Food Addiction (SPFA)

To assess whether or not participants believed themselves to be a food addict, participants were presented with the statement “I believe myself to be a food addict” with response options “Yes” or “No”. Similar measures have been used in previous research, and positive responses on this assessment have been associated with greater food reward, overeating [23,38], and fear of being stigmatized by others [22].

#### 2.3.5. Employability Questions

For consistency with the study’s cover story, seven items were included which assessed participants’ beliefs about Paulina’s employability (e.g., How likely would you be to employ Paulina?). Responses were recorded using Visual Analogue Scales (VAS) ranging from 0 (not at all) to 100 (extremely). Higher scores indicated more positive attitudes towards Paulina’s employability. Analyses of the effect of condition on employability ratings are presented in the Appendix A.

### 2.4. Data Analysis

A MANOVA was conducted to check whether participants differed between conditions on age, BMI, and DEBQ subscale scores. Chi-squared tests were conducted to check for any differences between the proportion of students/non-students and Caucasian/non-Caucasian participants allocated to each condition. To examine the effect of condition on target-specific and general stigma, two ANOVAs were conducted with the condition (i.e., control, medical, self-diagnosed) as the independent variable, and M-FPS (i.e., target specific stigma) and AFA (i.e., general stigma) scores as dependent variables. Where significant main effects were identified, these were followed up by inspecting pairwise comparisons.

We conducted exploratory analyses to examine whether self-reported BMI moderated the effect of condition on mean Modified Fat Phobia Scale (M-FPS) and Anti-Fat Attitudes (AFA) scores. To do this, we conducted two hierarchical multiple linear regression to examine the relative contributions of BMI (centered) and condition to mean M-FPS scores and AFA scores. All three conditions were dummy coded with the Control condition as the reference variable. To assign dummy codes, two dummy variables were created: *D*_1_ (Medical) and *D*_2_ (Self-diagnosed). Participants in the medical condition were assigned ‘1’ to *D*_1_, and ‘0’ for *D*_2_. Participants in the self-diagnosed condition were assigned ‘0’ to *D*_1_ and 1 to *D*_2_. Participants in the control condition (i.e., the reference category) were assigned 0 to both *D*_1_ and *D*_2._ (see [39] for more information about dummy coding). Dummy-coded conditions were then entered into Step 1 of each regression model, along with BMI. The interaction terms (i.e., BMI × medical vs. control/self-diagnosed vs. control) were entered into Step 2 of the model. 

Additional exploratory analyses were conducted to examine whether the effect of condition on target-specific and general stigma was moderated by participants’ age or DEBQ subscales. Further details and results from these analyses are provided in the Appendix A.

## 3. Results 

### 3.1. Participant Characteristics

The MANOVA revealed that BMI differed significantly between conditions, F(2,434) = 4.80, *p* = 0.009, ηp^2^ = 0.022. This was due to a higher mean BMI in the medical condition relative to the self-diagnosed condition (*p* = 0.002). Participant characteristics as a function of condition are displayed in Table 1. Participants did not differ with regards to age or scores on DEBQ-subscales. Chi-squared tests (X^2^) revealed no difference in the proportion of students/non-students and Caucasian/non-Caucasian participants in each condition.

### 3.2. Effect of Condition on Target-Specific and General Stigma

There was a main effect of condition on mean Modified Fat Phobia Scale (M-FPS) score (i.e., target-specific stigma), F(2,437) = 9.07, *p* < 0.001, ηp^2^ = 0.040. Pairwise comparisons revealed that, compared to those in the control condition, M-FPS scores were higher in the medical (*p* < 0.001) and self-diagnosed (*p* = 0.001) conditions (Figure 1) (Control condition: Mean = 3.47, SD = 0.47, range = 2.29–4.71; Self-diagnosed: Mean = 3.66, SD = 0.48, range = 2.71–4.93; Medical: Mean = 3.68, SD = 0.52, range = 1.00–5.00). There was no difference in mean M-FPS scores between those in the medical and self-diagnosed conditions (*p* = 0.730). There was no effect of condition on Anti-Fat Attitudes (AFA) total scores (i.e., general stigma), F(2,437) = 0.754, *p* = 0.471, (Control condition: Mean = 1.78, SD = 0.56, range = 0.31–3.46; Self-diagnosed: Mean = 1.71, SD = 0.56, range = 0.23–3.00; Medical: Mean = 1.72, SD = 0.56, range = 0.38–3.38).

### 3.3. Moderating Effect of BMI

Hierarchical linear regression analyses were conducted to examine whether BMI moderated the effect of condition on target-specific (i.e., M-FPS scores) and general (AFA scores) stigma. Results from the exploratory analysis predicting M-FPS scores are provided in Table 2. In Step 1 and Step 2 of the model, M-FPS scores were significantly predicted by both condition (medical vs. control and self-diagnosed vs. control) and BMI; higher BMI was associated with lower M-FPS scores. However, M-FPS scores were not significantly predicted by the BMI × Condition interaction terms in Step 2 of the model. 

Neither BMI nor condition predicted AFA scores in Step 1 of the model (r^2^ = 0.005, *p* = 0.510), and the inclusion of interaction terms in Step 2 did significantly improve the fit of the model r^2^ = 0.015, *p*= 0.124).

## 4. Interim Discussion

Study 1 found that female participants who were exposed to medical and self-diagnosed food addiction vignettes exhibited more target-specific stigma towards a woman with obesity than those in the control condition. This is consistent with previous research in which the food addiction label was found to exacerbate stigmatizing attitudes towards an individual with obesity and ‘food addiction’ [27]. 

One possibility is that ‘food addiction’ stigma may be particularly high amongst those who perceive addiction to be within personal control [7]. This is supported by previous research in which perceiving addiction as a disease, rather than due to personal choice, was associated with reduced stigma towards people with addictive disorders [40,41]. Similarly, biogenetic explanations have been found to reduce stigma towards obesity, problematic eating, and substance abuse, relative to behavior-based explanations [10,31,42]. In Study 2, we examined whether the effect of food addiction condition on stigma would be moderated by the extent that addiction is viewed as a ‘disease’ relative to personal choice.

We also examined whether stigmatizing attitudes towards the target with food-addiction would be moderated by individuals’ scores on a measure of addiction-like eating. Previous research has found that individuals with personal experience of addiction have less negative attitudes towards others with addiction [43]. Furthermore, social identity theory suggests that individuals view other ‘in-group’ members more favorably than out-group members [44]. Therefore, we predicted that the effect of condition on target-specific stigma would be attenuated in participants with greater levels of addiction-like eating behavior.

Finally, we examined whether the effect of condition on target-specific and general stigma would differ between males and females. Previous research has found that females demonstrate less obesity-related stigma and stigma towards the ‘food addiction’ label than males [27]. We, therefore, hypothesized that the exacerbating effect of the food addiction label on stigma would be most pronounced in males.

To summarize, Study 2 examined the following hypotheses: (1) The effect of condition on target-specific and general stigma would be attenuated in those with greater support for the disease model of addiction. (2) The effect of condition on stigma would be attenuated in those who score highly on a measure of addiction-like eating, relative to those who score lower on addiction-like eating. (3) The effect of condition on stigma would be attenuated in females, relative to males.

## 5. Study 2 Method

### 5.1. Participants

Male and female participants, aged over 18 years, were invited to take part in a study into ‘employability perceptions’. A total of 523 (190 males; 314 females; 19 did not disclose their gender) participants completed the study. Six hundred and ten participants started the online survey, but 87 either did not complete it or were aged under 18 years old and were excluded from analyses. Participants were recruited from the University of Liverpool (*n* = 333) and Newcastle University (*n* = 190) in the UK. The mean age of participants was 27.1 (SD = 11.3) years, and the mean self-reported BMI was 23.6 kg/m^2^ (SD = 4.1). Participants with self-reported BMI over 30 kg/m^2^ (i.e., classified as having obesity) comprised 7.1% of the sample, 21.6% had a self-reported BMI between 25–29.9 kg/m^2^ (i.e., ‘overweight’), 64.4% had a self-reported BMI between 18.5–24.9 kg/m^2^ (i.e., healthy weight), and 5.5% had a self-reported BMI below 18.5 kg/m^2^ (i.e., ‘underweight’). Just over half of the sample were university students (*n* = 275, 52.4%) and the majority were Caucasian (*n* = 465, 88.9%). Ethical approval was granted by the relevant ethics committee at each of the two sites (University of Liverpool approval code: IPHS-1516-SMc-259-Generic RETH000619; Newcastle University approval code 1485/4293).

### 5.2. Materials and Procedure

Study 2 used the same materials and procedure as Study 1 but with the following additional measures:

#### 5.2.1. Addiction Belief Scale (ABS)

The ABS [39] was used to measure beliefs about addiction. Nine items assessed the belief that addiction is a disease (disease subscale, Cronbach’s α = 0.590), and nine items assessed the belief that addiction is within personal control (free will subscale, Cronbach’s α = 0.546). Items were rated on a 5-point Likert scale ranging from ‘strongly disagree’ to ‘strongly agree’. Higher scores indicate greater support for the belief that addiction is akin to a disease (disease subscale), and a matter of personal choice (free will subscale).

#### 5.2.2. Addiction-Like Eating Behaviour Scale (AEBS)

The AEBS [45] consists of 15 items which assess the presence of behaviors that are commonly associated with addiction-like eating (e.g., ‘I continue to eat despite feeling full’). Responses are provided on 5-point Likert Scales ranging from ‘Strongly disagree’ to ‘Strongly agree’, and from ‘Never’ to ‘Always’. The scale comprises two subscales: appetitive drive (9 items, Cronbach’s α = 0.890) and low dietary control (6 items, Cronbach’s α = 0.806). Higher scores indicate greater addiction-like eating behavior. Previous research suggests that this measure correlates positively with other measures of disinhibited eating (i.e., the Binge Eating Scale, [46]) and explains greater variance in BMI over and above other measures of ‘food addiction’ such as the Yale Food Addiction Scale [47].

#### 5.2.3. Data Analysis

A MANOVA was conducted to check whether participants differed, between conditions, with regards to age, BMI, DEBQ subscales scores, and scores on the Addiction-like Eating Behaviour Scale (AEBS) and Addiction Belief Scale (ABS). Chi-squared tests were conducted to check for any differences between the proportion of students/non-students, Caucasian/non-Caucasian, and males/females allocated to each condition. As in Study 1, two univariate ANOVAs were conducted to examine the effect of condition on Anti-fat Attitudes (AFA; general stigma) and Modified-Fat Phobia Scale (M-FPS) scores (target-specific stigma). Gender was also included in the model as a between-subjects variable.

Hierarchical multiple linear regression analyses were conducted to examine whether any effects of condition on target-specific and general stigma were moderated by support for the ‘disease’ model of addiction (i.e., ABS-disease scores), and addiction-like eating behavior (i.e., AEBS scores). All three conditions were dummy coded with the Control condition as the reference variable. To assign dummy codes, two dummy variables were created: *D*_1_ (Medical) and *D*_2_ (Self-diagnosed). Participants in the medical condition were assigned ‘1’ to *D*_1_, and ‘0’ for *D*_2_. Participants in the self-diagnosed condition were assigned ‘0’ to *D*_1_ and 1 to *D*_2_. Participants in the control condition (i.e., the reference category) were assigned 0 to both *D*_1_ and *D*_2._ (see [48] for more information about dummy coding). Dummy-coded conditions were then entered into Step 1 of each regression model, along with Addiction Belief Scale (disease subscale) or AEBS scores. The interaction terms (i.e., AEBS/Addiction Belief Scale (disease subscale) × medical vs. control/self-diagnosed vs. control) were entered into Step 2 of the model. Separate regression analyses were conducted to examine the ability of each interaction term to predict AFA scores (i.e., general stigma) and M-FPS scores (i.e., target-specific stigma). Addiction Belief Scale (disease subscale) and AEBS scores were centered prior to analyses.

## 6. Results

### 6.1. Participant Characteristics

Participants did not differ between conditions on any of the assessed characteristics (Table 3).

### 6.2. Effect of Condition and Gender on Target Specific Stigma

In contrast to Study 1, there was no main effect of condition on target-specific stigma, F(2,517) = 0.69, *p* = 0.501, (Control condition: Mean = 3.56, SD = 0.48, range = 2.43–5.00; Self-diagnosed: Mean = 3.63, SD = 0.47, range = 2.36–4.64; Medical: Mean = 3.63, SD = 0.47, range = 2.57–4.93). Contrary to hypothesis 3, there was no gender × condition interaction for target-specific stigma, F(2,517) = 1.18, *p* = 0.309. However, there was a main effect of gender, F(1,517) = 5.13, *p* = 0.024, ηp^2^ = 0.010, such that males had significantly higher scores on the Modified Fat Phobia Scale (M-FPS) than females i.e., they showed higher levels of target-specific stigma (Males: M = 3.67, SE = 0.034; Females: M = 3.57, SE = 0.026).

### 6.3. Effect of Condition and Gender on General Stigma 

As in Study 1, there was no effect of condition on Anti-fat Attitudes (AFA) scores (i.e., general stigma), F(2,517) = 1.18, *p* = 0.308, (Control: Mean = 4.34, SD = 1.00, range = 2.15–7.31; Self-diagnosed: Mean = 4.17, SD = 1.00, range = 1.54–7.15; Medical: Mean = 4.29, SD = 1.09, range = 1.31–7.85). Contrary to hypothesis 3, there was no gender × condition interaction, F(2,517) = 0.02, *p* = 0.978. There was also no main effect of gender on AFA scores, F(1,517) = 0.02, *p* = 0.978. For further analyses of gender differences on the AFA subscales, please see the Appendix A.

### 6.4. Effect of Disease Beliefs on Stigma

Scores on the disease subscale of the Addiction Belief Scale (ABS) significantly predicted mean Modified-Fat Phobia Scale (M-FPS) scores in Step 1 and Step 2 of the model such that higher scores on the scale (i.e., greater belief that addiction is akin to a disease) were associated with greater target specific stigma (i.e., higher M-FPS scores) (Table 4). However, M-FPS scores were not significantly predicted by condition, and there was no condition × ABS-disease interaction, contrary to our hypothesis. Step 1: r = 0.204, r^2^ = 0.042, *p* < 0.001; Step 2: r = 0.204, r^2^ = 0.042, *p* = 0.972.

Similarly, scores on the disease subscale of the ABS significantly predicted Anti Fat Attitude (AFA) scores (i.e., general stigma) in Step 1 and Step 2 of the model such that higher scores on the ABS-disease subscale predicted higher AFA scores (Table 5). Contrary to hypothesis 1, AFA scores were not significantly predicted by condition, and there was no interaction between condition and disease scores on AFA.

### 6.5. Addiction-Like Eating Behavior

Addiction-like Eating Behavior Scale (AEBS) scores and condition did not predict Modified Fat Phobia Scale (M-FPS) (target-specific stigma) scores in Step 1 of the model. However, the inclusion of the interaction terms in Step 2 significantly improved the fit of the model. Regression coefficients revealed a significant interaction between AEBS scores and medical (vs. control) condition on M-FPS scores (Table 6).

To further examine the interaction between AEBS scores and condition on M-FPS scores, we used the Johnson–Neyman technique [49] to identify the levels of addiction-like eating (i.e., AEBS scores) at which condition elicited a significant difference on M-FPS scores [50]. Using PROCESS (Version 3.1., [51]), the Medical (dummy-coded) condition was entered as the predictor variable, AEBS scores were entered as the moderator variable, and Self-diagnosed condition (dummy-coded) and the Self-diagnosed × AEBS interaction term were entered as covariates. Mean-FPS scores were entered as the dependent variable. This analysis showed that the Medical condition resulted in significantly greater M-FPS scores, relative to the Self-diagnosed and Control conditions (ps < 0.05), but only for those with low AEBS scores (centered AEBS score ≤ –2.81) (Figure 2). Findings are, therefore, consistent with our hypothesis that the effect of condition on stigma would be attenuated in those with higher levels of addiction-like eating behavior.

The condition × AEBS scores model predicting (general stigma) AFA scores was not significant (Step 1: r = 0.069, r^2^ = 0.005, *p* = 0.484; Step 2: r = 0.084, r^2^ = 0.007, *p* = 0.540).

## 7. Discussion 

Across two studies, we examined the effect of the food addiction label on stigmatizing attitudes towards an individual with obesity (i.e., target specific), and towards people with obesity more generally (i.e., general stigma). In Study 1, participants in both the medical and self-diagnosed food addiction conditions demonstrated greater target-specific stigma relative to the control condition. There was no effect of condition on general stigmatizing attitudes towards people with obesity. However, findings from Study 1 were not replicated in Study 2, in which we included both male and female participants. That is, we found no overall differences between the food addiction conditions and the control condition on target-specific stigma. The effect of condition on target-specific or general stigma was also not moderated by addiction disease beliefs (i.e., the extent to which addiction is perceived as a disease) or gender, in Study 2. However, there was a significant condition by addiction-like eating behavior interaction on target-specific stigma; participants who scored low on a measure of addiction-like eating demonstrated greater target-specific stigma in the Medical condition relative to Control and Self-diagnosed conditions. In contrast, target-specific stigma did not differ as a function of condition for those with high levels of addiction-like eating.

Findings from Study 1 are consistent with previous findings in which the food addiction label added to the stigma of obesity [27]. Higher levels of stigma towards the ‘self-perceived’ food-addicted target in the current study may reflect perceptions of food addiction as an ‘excuse’ for overeating. This is supported by qualitative evidence that individuals with overweight or obesity may be reluctant to label themselves as food addicts due to concerns that this would be perceived as an ‘excuse’ for their weight [29].

We predicted that the medical condition might legitimize the concept of food addiction and thereby reduce weight-related stigma (i.e., by removing personal responsibility from the individual). However, contrary to our hypothesis, in Study 1, we found that target-specific stigma was also higher in the medical condition compared to the control condition and did not differ from levels observed in the self-diagnosed condition. This finding is inconsistent with predictions from attribution theory [7] in which undesirable behaviors that are perceived as beyond personal control are thought to elicit less stigma than those that are perceived as controllable. One possibility is that food addiction explanations increase stigma by inadvertently emphasizing the behavioral aspect of obesity. That is, food addiction may imply a loss of control over eating, and previous studies have found that this may increase stigmatizing attitudes towards obesity [52]. Another possible explanation is that food addiction, unlike other biological causes of obesity, is believed to be within personal control and that medicalizing the term does not remove perceptions of personal responsibility. Indeed, Lee et al. [21] reported that almost three-quarters of people supported food addiction as a cause of obesity, and yet obesity was still viewed as a condition that individuals need to take responsibility for. Therefore, it may be the case that stigmatizing attitudes towards ‘food addicted’ individuals are dependent upon the extent that addiction is perceived as being outside of personal control and/or akin to a disease. In relation to this, Study 2 examined whether the effect of food addiction condition on stigma would be attenuated in those with greater support for the disease model of addiction (results discussed below).

Study 1 therefore suggests that the food addiction label exacerbated stigmatizing attitudes towards a woman with obesity, regardless of whether the food addiction was medically diagnosed or self-diagnosed. Notably, findings from Study 1 are inconsistent with those obtained in a previous study in which a ‘food addiction’ explanation for obesity elicited *lower* levels of target-specific and general stigma than a control explanation [28]. This inconsistency may be attributable to the control conditions used in ours and Latner et al.’s [28] study; in the current study, participants in the control condition were not provided with any explanation for the target’s weight status. In contrast, participants in Latner et al.’s [28] study read that obesity is caused by repeatedly choosing to consume high-calorie foods. By emphasizing the role of personal choice, it is possible that the control condition used by Latner et al. [28] may have elicited greater stigma than a ‘food addiction’ explanation for obesity.

In Study 2, we found that greater support for the disease model of addiction was associated with greater target-specific and general stigma towards obesity. This finding was unexpected and is contrary to predictions derived from attribution theory. One possibility is that the perception of addiction as a ‘disease’ encourages the view that addicts are abnormal and perpetuates an ‘us-them’ distinction [53]. Holding disease views of addiction also suggests that the person’s condition is irrevocable and permanent [54]. Another possibility is that causal beliefs about food addiction do not coincide with perceptions of other addictions. That is, individuals who support the ‘disease’ model for substance-based addictions may not necessarily attribute food addiction to a disease. Previous research supports this, indicating that addictions vary in the extent to which they are attributed to disease or personal choice. In particular, de Pierre et al. [40] found that food addiction was perceived as less of a disease and more within personal control compared with other addictions such as alcoholism. The measure of addiction beliefs (i.e., the ABS) used in the current study referred to addiction in general, and thus may not have reflected participants’ beliefs about food addiction per se.

However, the moderating effect of addiction-like eating on target-specific stigma, observed in Study 2, suggest that medically diagnosed food addiction could exacerbate weight-related stigma but only for people with low levels of addiction-like eating tendencies. A possible explanation for this finding is that individuals with personal experience of problematic eating (i.e., high AEBS scores) may have identified more with the target in the vignette and thereby displayed less negative attitudes towards her food addiction (e.g., see [43,44]) as opposed to participants with low AEBS scores.

In Study 2, male participants demonstrated significantly higher target-specific stigma, relative to female participants. Males and females did not differ on a measure of general weight-related stigma. However, the lack of interaction between gender and condition is inconsistent with previous research [27] in which stigmatizing attitudes towards a ‘food addicted’ target were lower in females, relative to males. This null result may be explained by the fact that, in the current study, males had a significantly higher mean BMI than females (see Appendix A). A previous study found that people with higher BMI hold less stigmatizing attitudes towards the ‘food addict’ label, relative to those with lower BMI [27]. Consistent with this, in Study 1, we found that higher BMI was associated with lower target-specific weight stigma. It is therefore possible that, in the current study, any moderating effect of gender on stigma may have been masked by the higher BMI of male, relative to female, participants. Future research should examine the moderating effect of gender on stigmatizing attitudes towards a food-addicted target in samples of males and females matched for BMI.

The inconsistent findings obtained across Studies 1 and 2 could not be attributable to the inclusion of males in Study 2 as the effect of condition on target-specific stigma was not moderated by gender. The sample tested in Study 2 comprised a larger proportion of older, non-students than the sample tested in Study 1. However, exploratory analyses revealed that the effect of condition on stigma was not moderated by student status or age (see online Appendix A). Differences between Studies 1 and 2 are, therefore, likely due to another (unknown) variable. Moreover, these findings suggest that the effects of the food addiction label on weight-related stigma may not be generalizable across populations.

There are several limitations to the current study that require consideration. Firstly, we note that the Addiction Belief Scale, used in Study 2, examined beliefs about the causes of addiction in general, and thus may not have captured individual differences in beliefs about the causes of food addiction. Future research could use an adapted version of the ABS (such as that used by de Pierre et al. [40]) to test whether food addiction stigma is attenuated in individuals who have greater support for a disease model of food addiction. Secondly, we did not examine whether participants believed the food addiction explanation for obesity, nor did we check whether participants had guessed the study aims. It is, therefore, possible that the effect of the food addiction label on stigma, observed in Study 1, could be due to demand characteristics that were not present in Study 2. Thirdly, the use of a female target in the current study precludes the generalizability of our findings to males. Previous research suggests that females are more likely than males to be stigmatized due to their weight [55], and so attitudes towards the food addiction label may similarly differ as a function of the target’s gender. Finally, it is important to consider that the findings may have been affected by the order in which the questionnaires were presented. In particular, the significant effect of condition on target-specific stigma (M-FPS) (in Study 1), and lack of effect of general stigma (AFA), may be due to the fact that participants completed the M-FPS immediately after reading the vignette, while general stigma (i.e., AFA scores) were assessed later in the study.

Future research should aim to clarify the effect of the food addiction label on weight-related stigma. This may be achieved by considering possible moderating effects of pre-existing beliefs about food addiction (e.g., the extent that it is a legitimate condition, whether it is controllable, etc.). There has been much debate in the scientific literature about whether addiction-like eating should be considered a substance-based ‘food addiction’ or a behavioral ‘eating addiction’ (e.g., [11]). Therefore, it will also be important to compare attitudes elicited by a ‘food addiction’ label, with attitudes towards an ‘eating addiction’ label. It would also be interesting to compare the effect on the stigma of medically-diagnosed food addiction, with other medical causes of weight gain (e.g., hypothyroidism). Doing so would provide insight into whether the potential exacerbating effect of medicalization on stigma is specific to the food addiction label or whether it extends to the medical model per se. It is also possible that emphasizing the non-behavioral aspect of food addiction (e.g., brain differences to food) may reduce any deleterious effect of a medical diagnosis on stigma. More broadly, the clinical implications of food addiction labels on weight-related stigma must now be considered. In particular, it is important to consider whether the food addiction label may affect people’s approaches to treatment (e.g., seeking pharmacological solutions rather than psychotherapy). It is also possible that, by perpetuating weight-related stigma, the food addiction label could be detrimental to psychological well-being and undermine people’s attempts to lose weight.

## 8. Conclusions

The results indicate that the food addiction label may exacerbate stigmatizing attitudes towards an individual with obesity. Furthermore, there is preliminary evidence that this effect may be most pronounced in people with pre-existing low levels of addiction-like eating behavior. Further research is needed to determine the longer-term effects of the food addiction label on weight stigma and clinical implications.

## Figures and Tables

**Figure 1 nutrients-11-02100-f001:**
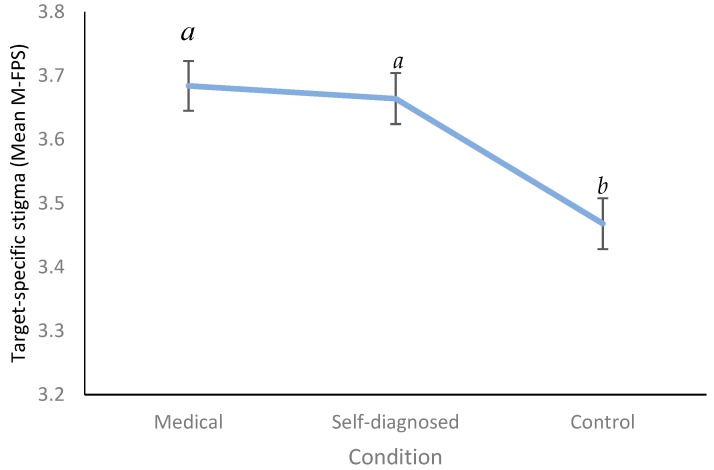
Mean Modified Fat Phobia Scale (M-FPS) scores (i.e., target specific stigma) as a function of condition. Different letters indicate significant differences. Higher scores indicate more negative attitudes towards Paulina (i.e., higher levels of target-specific stigma). Error bars denote standard error.

**Figure 2 nutrients-11-02100-f002:**
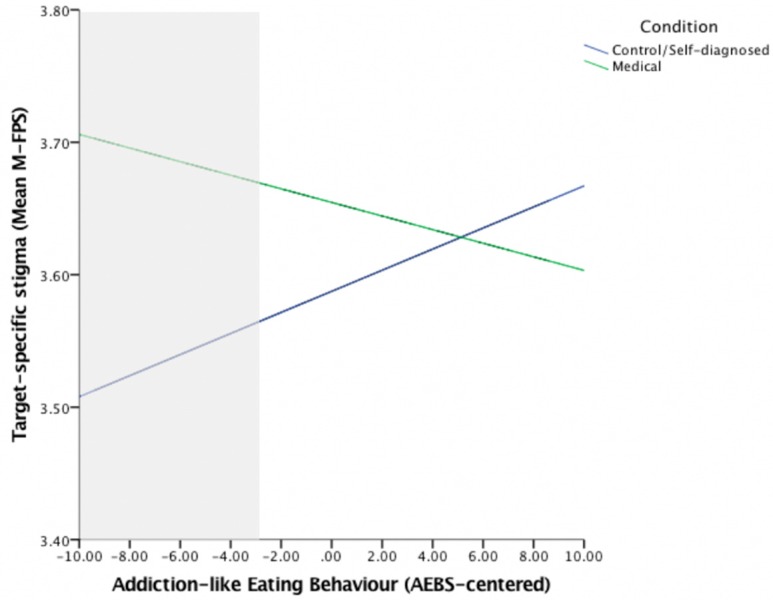
The effect of condition on M-FPS scores at different levels of addiction-like eating behavior (assessed using the AEBS). The shaded area represents the region of significance identified using the Johnson-Neyman technique.

**Table 1 nutrients-11-02100-t001:** Participant characteristics as a function of condition.

Variable	Medical (*N* = 148)	Self-Diagnosed (*N* = 144)	Control (*N* = 146)	Between-Group Differences
Age (y)	21.09 (±6.44)	21.07 (±7.45)	21.38 (±7.32)	F(2,435) =0.09, *p* = 0.916
BMI (kg/m^2^)	22.60 (±3.22) *	21.60 (±2.95) *	22.04 (±2.93)	F(2,432) =3.64, *p* = 0.027
DEBQ-Restraint	2.94 (±0.96)	2.72 (±0.90)	2.89 (±0.86)	F(2,436) =2.38, *p* = 0.094
DEBQ-Emotion	2.97 (±0.90)	2.80 (±0.90)	2.84 (±0.86)	F(2,436) =1.43, *p* = 0.240
DEBQ-External	3.34 (±0.69)	3.21 (±0.59)	3.35 (±0.71)	F(2,436) =1.95, *p* = 0.143
Ethnicity (% Caucasian)	93.3	91.0	86.4	X^2^(2) = 4.12, *p* = 0.127
Occupation (% students)	83.2	83.3	81.6	X^2^(2) = 0.186, *p* = 0.911

Results are means (standard deviations) unless otherwise specified (* significant difference, *p* < 0.05).

**Table 2 nutrients-11-02100-t002:** Regression output with mean M-FPS (i.e., target-specific stigma) as the dependent variable.

Model	B	SE	*t*	*p*
**Step 1**				
BMI	−0.015 *	0.007	−2.119	0.035
Medical	0.230 **	0.056	4.109	<0.001
Self-diagnosed	0.189 **	0.056	3.360	0.001
**Step 2**				
BMI	−0.034 *	0.013	−2.547	0.011
Medical	0.223 **	0.056	3.992	<0.001
Self-diagnosed	0.190 **	0.056	3.360	0.001
BMI × Medical	0.031	0.017	1.816	0.070
BMI × Self-diagnosed	0.019	0.019	0.980	0.327

* *p* < 0.05, ** *p* < 0.01. Step 1: r^2^ = 0.051, *p* < 0.001; Step 2: r^2^ = 0.058, *p* = 0.194

**Table 3 nutrients-11-02100-t003:** Participant characteristics as a function of condition (Study 2).

Variable	Medical (*N* = 178)	Self-Diagnosed (*N* = 175)	Control (*N* = 170)	Between-Group Differences
Age (y)	26.6 (11.1)	26.9 (10.9)	27.8 (12.0)	F(2,511) = 0.34, *p* = 0.711
BMI (kg/m^2^)	23.6 (4.5)	23.6 (4.2)	23.5 (3.7)	F(2,511) = 0.03, *p* = 0.974
DEBQ-Restraint	2.66 (0.91)	2.67 (.86)	2.76 (0.90)	F(2,511) = 0.47, *p* = 0.626
DEBQ-Emotion	2.67 (0.90)	2.64 (.98)	2.77 (0.99)	F(2,511) = 1.16, *p* = 0.314
DEBQ-External	3.29 (0.58)	3.26 (.57)	3.38 (0.55)	F(2,511) = 2.44, *p* = 0.088
AEBS	36.57 (9.65)	35.99 (9.87)	36.05 (8.70)	F(2,511) = 0.33, *p* = 0.720
ABS-disease	25.80 (3.75)	25.19 (3.92)	25.86 (4.41)	F(2,511) = 1.45, *p* = 0.236
ABS-Free Will	30.01 (3.29)	29.95 (3.72)	30.15 (4.04)	F(2,511) = 0.14, *p* = 0.873
Ethnicity (% Caucasian)	89%	89%	88%	X^2^(2) = 0.119, *p*=0.942
Occupation (% students)	57%	49%	52%	X^2^(2) = 2.08, *p* = 0.354
Gender (% male)	42%	31%	38%	X^2^(2) = 4.95, *p* = 0.084

Abbreviations: AEBS, Addiction-like Eating Behavior Scale; ABS, Addiction Beliefs Scale; DEBQ, Dutch Eating Behaviour Scale.

**Table 4 nutrients-11-02100-t004:** Regression output for Addiction Belief Scale (ABS)-disease with M-FPS (target-specific stigma) as the dependent variable.

Model	B	SE	*t*	*p*
**Step 1**				
Medical	0.072	0.050	1.427	0.154
Self-diagnosed	0.091	0.051	1.797	0.073
ABS-disease	0.023 **	0.005	4.439	0.000
**Step 2**				
Medical	0.071	0.050	1.415	0.158
Self-diagnosed	0.090	0.051	1.781	0.076
ABS-disease	0.022 **	0.008	2.685	0.007
ABS-Disease × Medical	0.002	0.012	0.195	0.846
ABS-Disease × Self-diagnosed	0.000	0.012	−0.034	0.972

** *p* < 0.01. The control condition was used as the reference category against which medical and self-diagnosed conditions were compared. Abbreviations: ABS, Addiction Belief Scale. Step 1: r^2^ = 0.042, *p* < 0.001; Step 2: r^2^ = 0.042, *p* = 0.972).

**Table 5 nutrients-11-02100-t005:** Regression output for ABS-disease with Anti Fat Attitude (AFA; general stigma) as the dependent variable.

Model	B	SE	*t*	*p*
**Step 1**				
Medical	−0.056	0.109	−0.516	0.606
Self-diagnosed	−0.146	0.110	−1.331	0.184
ABS-disease	0.047 **	0.011	4.281	0.000
**Step 2**				
Medical	−0.053	0.109	−0.482	0.630
Self-diagnosed	−0.147	0.110	−1.337	0.182
ABS-disease	0.059 **	0.018	3.295	0.001
ABS-Disease × Medical	−0.016	0.027	−0.582	0.560
ABS-Disease × Self-diagnosed	−0.021	0.026	−0.791	0.429

** *p* < 0.01. The control condition was used as the reference category against which medical and self-diagnosed conditions were compared. Step 1: r = 0.198, r^2^ = 0.039, *p* < 0.001; Step 2: r = 0.201, r^2^ = 0.040, *p* = 0.707.

**Table 6 nutrients-11-02100-t006:** Regression output for Addiction-like Eating Behavior Scale (AEBS) scores with M-FPS (target-specific stigma) as the dependent variable.

Model	B	SE	*t*	*p*
**Step 1**				
Medical	0.067	0.051	1.32	0.186
Self-diagnosed	0.071	0.051	1.38	0.168
AEBS	0.003	0.002	1.42	0.156
**Step 2**				
Medical	0.067	0.051	1.32	0.187
Self-diagnosed	0.071	0.051	1.39	0.165
AEBS	0.008	0.004	1.90	0.058
AEBS × Medical	−0.013 *	0.006	−2.35	0.019
AEBS × Self-diagnosed	0.000	0.006	−0.065	0.948

* *p* < 0.05. The control condition was used as the reference category against which medical and self-diagnosed conditions were compared. Abbreviations: AEBS, Addiction-like Eating Behavior Scale. Step 1: r^2^ = 0.009, *p* = 0.214; Step 2: r^2^ = 0.023, *p* = 0.020.

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
