# Peer review of "Obesity Stigma: Is the ‘Food Addiction’ Label Feeding the Problem?"

_nutrients, 2019, doi:10.3390/nu11092100_

Round 1

Reviewer 1 Report

Nutrients ID nutrients-569810: Obesity Stigma: Is the ‘Food Addiction” Label Feeding the Problem?

This manuscript describes two studies. The studies provided participants with a picture of a woman with excess weight and a vignette to examine how the label of “food addiction” impacted target-specific stigma, weight-based stigma, and causal beliefs about addiction. In both studies, there were three conditions: 1) the target had been diagnosed with food addiction by her doctor (“medical” condition), 2) the target labeled herself as having food addiction (“self-diagnosed” condition”), and 3) a “control” condition, with no reference to food addiction. In the first study, 439 participants completed the study procedure, and participants in the medical and self-diagnosed conditions had greater target-specific stigma relative to the control condition; however, there was no impact on general weight-based stigma. In the second study, 523 participants completed the study procedure. There was no main effect of condition on target-specific or weight-based stigma; however, there was an interaction between condition and “addiction-like eating behaviour”, such that participants in the medical condition with low addiction-like eating behaviours had greater target-specific stigma than the control condition. The authors conclude that the food addiction label might increase stigmatizing attitudes towards that person, particularly if the individual has low levels of addiction like eating behaviours.   

Overall, this study is a solid contribution to the literature. I think it is important to better understand how labels such as “food addict” can impact people’s perceptions of the individuals with these labels, as well as broader attitudes towards individuals with excess weight. However, I did find the manuscript hard to follow at times. I think the manuscript could be improved with significant updates to the Data Analytic Plan and Results sections, as well as a more streamlined joint Discussion section across the two studies. Specific concerns and suggestions by section are as follows:

Abstract:

If true, it may be helpful to state that these two samples were separate and no participants enrolled in both studies. I was unsure what “target-specific stigma” was referring to until the Methods section- it may be helpful for the reader to further explain this construct in the Abstract and/or Introduction.

Introduction:

Overall, I thought the Introduction was well-written. However, I think a little bit more information on the background studies would be helpful when interpreting the study methodology and findings: Page 2, lines 48-50: what was the control condition? Page 2, lines 67-70: what do the authors mean by “despite strong support for the food addiction model of obesity”? Were the participants given a statement of the evidence supporting this model? Did this study have a control condition?

Study 1 & 2 Methods:

It may be helpful to move the Procedure section between the Participants and Measures section, and also to note explicitly if the order listed is the order participants completed the measures (or if measure order was randomized). Can the authors provide more information on the validity of the questionnaires? For example, what has the M-FPS been associated with in previous studies? Does it seem to predict outcomes? Is the DEBQ associated with differential eating behavior in the laboratory, for example? Has the AEBS been associated with other measures of disinhibited eating or just BMI? For those who are not familiar with these specific measures, it would improve readability in abbreviations were minimized throughout or the name of the constructs was used throughout the Results and Discussion sections- there were a large number of acronyms to try to remember while also trying to digest the findings. How did the authors use the employability questions? Did they sum the VAS ratings across the seven items?

Studies 1 & 2 Data Analytic Approach:

Studies 1 & 2: It would be helpful to add information about the post-hoc analyses (for the MANOVA and the ANOVAs) to the Data Analysis section. Study 1: The authors conducted exploratory analyses that they present in the Results section but do not mention here. The information about the analyses in section 3.3 (p. 5, line 199 onward) should be moved to the Data Analysis section. Study 1: The authors conducted exploratory analyses examining if BMI moderated the effects; why not also age and DEBQ scores? I am guessing it may be because only BMI was significant across groups. However, even if the mean is the same across groups, I believe there can still be a differential association between condition and moderator on the dependent variable. Study 1: Similarly, the authors only examined moderator effects of M-PFS (I assume because that was the only significant main effect); however, you can still conduct moderation analyses with non-significant main effects. In fact, if you do not find a main effect, it may be due to cross-over moderation effects. When examining some effects (e.g., gender moderating condition), why did the authors choose not to adjust for potential confounds, such as BMI? The authors even discuss that is a possible confound in the Discussion section (p. 12, lines 448-459).

Studies 1 & 2 Results

The authors do not present any data on the employability questions- did they find any effect of condition in either study? Can the authors provide the mean, standard deviation, and ranges of the outcome variables in both studies? For the exploratory moderating analyses in Study 1, I believe the authors should specify it is a hierarchical multiple linear regression (p. 5, line 201); same with the moderation analyses in Study 2 (pp. 7-8). Study 1: When examining moderating effects, I do not think it is appropriate to collapse across the two experimental conditions just because there was no main effect of condition on M-FPS; there may still be differential moderating effects. Study 1: The authors state “… when the interaction term was entered in Step 2 of the model, BMI predicted no additional variance in M-FPS scores…”, I would change this wording to state, “The main effect of BMI became non-significant” to more closely align with what you are testing (p. 5, line 207-p. 6, line 209).

Discussion and Conclusion:

I think it would have been interesting to include a condition in which the biological basis of excess weight was more readily accepted and less under someone’s behavioral control- for example, “Her doctor has diagnosed her with hypothyroidism”- this may help explain whether people are not identifying food addiction as a valid medical diagnosis, or whether even fully biological explanations increase target-specific stigma. This may be a useful future direction. I think future research should also consider directly assessing: How does the label of “food addiction” impact people’s perception of the cause and treatment of excess weight? These are potentially important clinical implications- for example, will people be more likely to see out pharmacological solutions instead of psychotherapy? Would adding etiological explanations (e.g., describing food addiction as neurotransmitter-driven for the medical vignette) change these findings, instead of just stating that her doctor diagnosed her with food addiction? There is already an Interim Discussion between Studies 1 and 2- I think a more integrated joint Discussion section is needed instead of having two separate Discussion sections; this way, the authors can consolidate their findings and interpret them together for the reader. Could the order of questionnaires have impacted findings?

Tables & Figures:

Figures: Figure 1: I think the letters on the figure may be incorrect; typically, significantly different groups are indicated by different letters, and non-significant differences by the same letter. So I think the medical and self-diagnosed conditions should be labeled “a”, and the control condition “b”, to indicate the first two conditions were similar and the last one was different from the other two. Same comment on Figure S1. Tables: Tables 2, 4, 5, 6: can the authors provide change R2/change in R2 to the table?

Minor comments:

On p. 5, line 190, the “p” in partial eta squared should be subscript. The authors may want to report the p-values to two decimal places to be consistent with other numbers provided.

Reviewer 2 Report

This topic of this paper is timely and worthwhile given the mixed research findings on "food addiction" and the potential clinical, societal and individual impacts that stem from the label.

The authors are to be commended for creating the two well-constructed studies presented in this paper. The writing is clear and well-organized and statistical analyses are appropriate. 

I have 2 minor points of feedback:

1. It appears that BMI was calculated from self-report information provided by participants. Please specify this, as there is some research that suggests that self-reported weight is often lower than measured weight.

2. The link to the supplementary materials at: www.mdpi.com/xxx/s1 appears to be broken.

Reviewer 3 Report

This is a very interesting vignette study exploring the effect of the "food addiction" explanatory model of obesity on stigma for this condition. It would have been interesting to add a condition where a medical explanatory model were given different from the addiction one (e.g.: appetite dysregulation) to test how much the "addiction" label more than the medical explanatory model per se is stigmatizing. This should probably be mentioned in the limitations.

Overall nice and well written study

Round 2

Reviewer 1 Report

The authors have addressed all my points sufficiently; I think the manuscript is much improved and would be appropriate for publication.